# *M*-ary Rank Classifier Combination: A Binary Linear Programming Problem

**DOI:** 10.3390/e21050440

**Published:** 2019-04-26

**Authors:** Vincent Vigneron, Hichem Maaref

**Affiliations:** Informatique, Bio-informatique et Systèmes Complexes (IBISC) EA 4526, univ Evry, Université Paris-Saclay, 40 rue du Pelvoux, 91020 Evry, France

**Keywords:** classifier combination, rank, aggregation, total order, independence, data fusion, mutual information, plurality voting, binary linear programming, cervical cancer, HPV

## Abstract

The goal of classifier combination can be briefly stated as combining the decisions of individual classifiers to obtain a better classifier. In this paper, we propose a method based on the combination of weak rank classifiers because rankings contain more information than unique choices for a many-class problem. The problem of combining the decisions of more than one classifier with raw outputs in the form of candidate class rankings is considered and formulated as a general discrete optimization problem with an objective function based on the distance between the data and the consensus decision. This formulation uses certain performance statistics about the joint behavior of the ensemble of classifiers. Assuming that each classifier produces a ranking list of classes, an initial approach leads to a binary linear programming problem with a simple and global optimum solution. The consensus function can be considered as a mapping from a set of individual rankings to a combined ranking, leading to the most relevant decision. We also propose an information measure that quantifies the degree of consensus between the classifiers to assess the strength of the combination rule that is used. It is easy to implement and does not require any training. The main conclusion is that the classification rate is strongly improved by combining rank classifiers globally. The proposed algorithm is tested on real cytology image data to detect cervical cancer.

## 1. Introduction

Using a single classifier has shown limitations in achieving satisfactory recognition performance, and this leads us to use multiple classifiers, which is now a common practice in machine learning. Classifier combination has been studied in many disciplines such as the social sciences, sensor fusion, pattern recognition, etc. Schapire [1] proved that a strong classifier can be generated by combining weak classifiers. It has been accepted as an effective method to improve classification performances. Many examples of ensemble classifier systems can be found in process engineering or medicine. For a survey of the issues and approaches on classifier combination, readers are referred to Woźniak [2] and Oza and Turner [3]. The same type of approach has also been used, for instance, in remote sensing domains (e.g., for land cover mapping with Landsat Multispectral Scanner, elevation) [4], computer security [5], financial risks [6], proteomics [7].

Classifiers can provide as their final decision only a single class, a ranked list of all the classes, or a score associated with each class as a measure of confidence for the class. In this paper, we focus only on rank-values to perform combination. Rank data are useful when data can not be easily reduced to numbers, such as data that are related to concepts, opinions, feelings, values, and behaviors of people in a social context, genes, characters, etc. Ranking also has the advantage of removing scale effects while permitting ranking patterns to be compared. But rank-ordering has also its disadvantages: it is difficult to combine data from different rankings, and the information contained in the data is limited [8].

After learning, each classifier of the ensemble has output its own results. Several fusion strategies have been proposed in the literature to combine classifiers at the rank level [9,10]. Among them, one of the most common techniques is certainly the linear combination of the classifier outputs [11,12]. The voting principle is the simplest method of combination, where the top candidate from each classifier constitutes a single vote. The final decisions can be made by majority rule (over half of the votes) [13], plurality (maximum number of votes) [14], weighted sum of significance [15], or other variants. The method of Borda count [16], which sums up the rank values of classifiers, can be considered as a generalization of the voting principle. The Bayesian approach estimates the class posterior probabilities conditioned on classifier decisions by approximating various probability densities [17]. Although decision theory itself does not assume classifiers are independent, this assumption is almost always adopted in practical implementation to reduce the exponential complexity of probability estimation. In summary, classifier combination is an ensemble method that classifies new data by taking a weighted vote of the predictions of a set of classifiers [18]. This is originally a Bayesian averaging, but more recent algorithms include boosting, bagging, random forests, and variants [19,20,21]. Note that Dempster–Shafer formalism for aggregating beliefs based on uncertainty reasoning lends itself to a more flexible model used to combine multiple pieces of evidence and capable of taking uncertainty and ignorance into account [22].

Finally, a rank classifier provides an ordered list of classes associating each class with a rank integer that indicates its importance in the list. The output of a classifier $K_{k}$ is therefore a vector of ranks attributed to the *K* classes.

An ensemble of classifiers might be a better choice than a single classifier because of the variability of the ensemble errors, which is such that the consensus performance is significantly better than the best individual in the ensemble [23]. This analysis is certainly true when the classifiers of the ensemble “see” different training patterns, and it can be effective even when the classifiers all share the same training set. In a computerized tomography problem to illustrate how the ensemble consensus outperformed the best individuals, Anthimopoulos observed that the marginal benefit obtained by increasing the ensemble size is usually low due to correlation among errors: most classifiers will get the right answer on easy inputs, while many classifiers will make mistakes on difficult inputs [24].

In addition, running several searches and combining the solutions produces a better approximation than many learning techniques that use local searches to converge toward a solution, with the risk of staying stacked in local optima (which may not be true in the case of deep learning classifiers, since Kawaguchi has shown that every local minimum is a global minimum [25]). Thus, we might not be capable of producing the optimal classifier using a training set and a given classifier architecture, compared to a set of several classifiers. Since the number of classifiers can be very high (in the thousands), it is difficult to “understand” the classifier ensemble decision characteristics.

Although general performances are often improved when classifiers are combined, it becomes computationally costly to combine well-trained classifiers [26]. Most of the time, it is believed that the combination of independent classifiers will provide greater performance improvement [27], while combiner decisions could be biased toward duplicated outputs. However, this belief stems from the difficulty of using a dependence assumption. In fact, in practical situations, classifier independence is difficult to assess.

How do multiple rank classifiers improve separation performances when individual classification performances are slightly better than random decision making? And what is “classifier independence” ? This term raises several issues that we will address in Section 4, where we come back to the theory of rank aggregation and propose an algorithm to combine classifiers. The main properties of the classifier are discussed. Section 2 exposes the general framework and the notations used. A classifier ensemble dependence measure is then proposed to evaluate the conditional mutual information in Section 5. Experimental results are presented in Section 6 for the detection of cervical cancer. Finally, Section 7 gives conclusions on rank classifier combination and further investigations are discussed.

Notations

Set and regions are indicated by double-trace uppercase letters such as G,S,R, vectors with bold lowercase such as x,y, and matrices with uppercase bold letters such as C,M,Σ. The elements of a matrix $M=\{m_{ij}\}$ are indexed by the row index *i* and the column index *j*. Lowercase letters refer to individual elements in a vector whose position in the vector is indicated by the last subscript. Therefore, $x_{ij}$ refers to the *j*th element of vector $x_{i}$. $p(C_{i})$ is the a priori probability of the random value *X* belonging to class $C_{i},1\leq i\leq K$, *K* being the number of classes. *M* is the number of classifiers used for combination. |C| denotes the cardinality of set C. ${}^{T}$ denotes the transpose operator.

## 2. Problem Statement and Model

We consider a classification dataset $B_{0}$ with *n* observations
(1)$$B_{0}={\left\{\left(x_{i},c_{i}\right)\right\}}_{i=1}^{n},$$
obtained from a physical signal, or synonymously, explanatory variables, objects, instances, cases, patterns, t-uples, etc. where each $x_{i}$ belongs to class $c_{i}\in \left\{C_{1},\cdots ,C_{K}\right\}$. The vector $x_{i}$ lies in an attribute space $A\in R^{p}$ and each component $x_{ij}$ is a numerical or nominal categorical attribute, also named feature, variable, dimension, component, field, etc.

The output of the *M* classifiers $K_{1}\left(x\right),\cdots ,K_{M}\left(x\right)$ are represented by a *K*-dimensional vector $u_{i}\left(x\right)={\left(u_{i1},\cdots ,u_{iK}\right)}^{T},1\leq i\leq m$: each component $u_{ij}$ is a certain value associated with class $C_{j}$ given by $K_{i}\left(x\right)$. Depending on the nature of the classifier $K_{i}\left(x\right)$, $u_{ij}$ can be a rank value that reflects a complete or partial ordering of all classes, or a value in {0,1} corresponding to the predicted class assigned to 1 and the others to zero, or a score, e.g., a discriminant value, associated with each class $C_{j}$, which serves as a confidence measure for the class to be the true class. The latter can easily be converted into the two former. Therefore, each classifier $K_{i}\left(x\right)$ defines a mapping function from the image domain $R^{p}$ to a *K*-dimensional vector space defined over a set of values $E_{i}$. The general framework is illustrated in Figure 1.

In this paper, $u_{ij}$ is a rank value that reflects a complete or partial ordering of the classes. The objective is to design an optimal combination function G that takes all the $u_{i}$ as input and produces as an output the decision vector $z={\left(z_{1},\cdots ,z_{K}\right)}^{T}$, where $z_{k}$ is the rank associated with the decision on class $C_{k}$, that is, $z=G(u_{1},\cdots ,u_{M})$. Thus, we seek G as a discriminant function defined over $R^{K\times m}$.

In the following, it is assumed that (*i*) classifiers have equal individual performance (*ii*) classifiers $K_{i}$ are treated as “black boxes”. Hence, the combination operator applies only on the real space vectors $u_{i}$.

## 3. Conditional Independence Properties

The term “classifier independence” has been used in an intuitive manner, but what is classifier independence? Formally, two classifiers $K_{1}$ and $K_{2}$ are said to be independent if
(2)$$p\left(u_{1}=c_{j},u_{2}=c_{\ell }\right)=p\left(u_{1}=c_{j},\right)p\left(u_{2}=c_{\ell }\right)\;\forall 1\leq j,\ell \leq K,$$
with $u_{1}$ and $u_{2}$ being the decision values of $K_{1}$ and $K_{2}$. The idea is illustrated in the following example.

**Example** **1**(Independent classifiers)**.**
*Consider a binary classification problem (with equiprobable classes $C_{1}$ and $C_{2}$) and two classifiers $C_{1}$ and $C_{2}$ with similar performances and whose outputs are $u_{1}$ and $u_{2}$, i.e., their probabilities of correct classification $\alpha_{1}$ and $\alpha_{2}$ are equal:*
(3)$$\begin{array}{cc} p(u_{1}=c_{1}|c_{1}) & =p\left(u_{1}=c_{2}|c_{2}\right)=\alpha_{1}\\ p(u_{2}=c_{1}|c_{1}) & =p\left(u_{2}=c_{2}|c_{2}\right)=\alpha_{2}\\ p(u_{1}=c_{1}|c_{2}) & =p\left(u_{1}=c_{2}|c_{1}\right)=1-\alpha_{1}\\ p(u_{2}=c_{1}|c_{2}) & =p\left(u_{2}=c_{2}|c_{1}\right)=1-\alpha_{2}. \end{array}$$*Then the total probability rule helps to find the probability of the outputs:*
(4)$$\begin{array}{cc} p(u_{1}=c_{1})= & p\left(u_{1}=c_{1}|c_{1}\right)p\left(c_{1}\right)+p\left(u_{1}=c_{1}|c_{2}\right)p\left(c_{2}\right)=\alpha_{1}/2+\left(1-\alpha_{1}\right)/2=1/2\\ p(u_{2}=c_{1})= & p\left(u_{2}=c_{1}|c_{1}\right)p\left(c_{1}\right)+p\left(u_{2}=c_{1}|c_{2}\right)p\left(c_{2}\right)=\alpha_{2}/2+\left(1-\alpha_{2}\right)/2=1/2. \end{array}$$*The two classifiers are independent if the joint probability $p(u_{1},u_{2})$ factorizes*
(5)$$p\left(u_{1}=c_{1},u_{2}=c_{1}\right)=p\left(u_{1}=c_{1}\right)p\left(u_{2}=c_{1}\right)=l\frac{1}{2}\times \frac{1}{2}=\frac{1}{4}.$$*And similarly,*
(6)$$p\left(u_{1}=c_{1},u_{2}=c_{2}\right)=p\left(u_{1}=c_{2},u_{2}=c_{1}\right)=p\left(u_{1}=c_{1}\right)p\left(u_{2}=c_{2}\right)=\frac{1}{2}\times \frac{1}{2}=\frac{1}{4}.$$In Equations (Equation 5)–(Equation 6), $\alpha_{1}$ and $\alpha_{2}$ do not appear anymore. The value of $p(u_{1},u_{2})$ should be $\frac{1}{4}$, independently of the classifier performances. This is possible only if $\alpha_{1}=\alpha_{2}=\frac{1}{2}$. Thus, the ensemble performance does not depend of the performance of the individuals. In other words, independent classifiers in the sense of definition (Equation 2) are random classifiers (recognition rate of 50%)!*Suppose now that classifiers are very efficient and that $\alpha_{1}$ and $\alpha_{2}$ are almost identical to 1. In this case, the probability that the two answers are correct is also almost equal to 1 and*
(7)$$p\left(u_{1}=c_{1},u_{2}=c_{2}\right)\approx p\left(c_{1}\right)=1/2\neq p\left(u_{1}=c_{1}\right)p\left(u_{2}=c_{2}\right),$$
*which is far from the value of $\frac{1}{4}$ required by the condition of independence.*

Example (1) suggests that interesting classifiers (non-random!) cannot be independent in the sense of Equation (Equation 2). Making the assumption that decision vectors $u_{1},\cdots ,u_{M}$ are conditionally independent given $x\in C_{j}$, the discriminant function G maximizes the posterior probability $p\left(C_{j}\right)\prod_{i=1}^{M}p\left(u_{i}|C_{j}\right)=p\left(C_{j}\right)\prod_{i=1}^{M}\prod_{k=1}^{K}p\left(u_{ik}|C_{j}\right)$, which can be point estimated from the entries of the *M*
KK-confusion matrices, as given, for instance, in Table 1.

Let ${𝟙}_{j<k}$ be the indicatrix function for which ${𝟙}_{j<k}=1$ if the rank of the class $C_{j}$ is less than the alternative class $C_{k}$, and 0 otherwise. Then in Table 1, $n_{jk}={𝟙}_{j<k}$ and the line and column marginals are respectively defined by $n_{j\;\cdot \;}=\sum_{k=1}^{K}n_{jk}$ and $n_{\;\cdot \;k}=\sum_{k=1}^{K}n_{jk}$. If class $C_{j}$ is the *k*th choice for classifier $K_{i}$, then $p\left(u_{ik}|C_{j}\right)=\frac{n_{jk}}{n_{j\;\cdot \;}}$.

**Example** **2**(Conditional independent classifiers)**.**
*Consider once again the binary classification case introduced in Example (1) and assume that the classifiers are very efficient: $\alpha_{1}=\alpha_{2}\approx 1$. Then*
(8)$$\begin{array}{cc} p(u_{1}=c_{2},u_{2}=c_{1}|c_{1}) & \approx 1\\ p\left(u_{1}=c_{1}|c_{1}\right)p\left(u_{2}=c_{1}|c_{1}\right) & =\alpha_{1}\alpha_{2}\approx 1 \end{array}$$
*We conclude that two classifiers can be conditionally independent even if they are very efficient. Equation (Equation 8) does not indicate that the classifiers are independent. It only suggests that they can be conditionally independent* or *conditionally dependent.*

Therefore, conditional independence can be seen as a necessary condition for classifier combination. But the direct use of the confusion matrix as a criterion to derive the optimal combination rule is not feasible since the true classes are unknown.

## 4. Rank Class Combination Problem

### 4.1. Rank-Order Statistic Model

A rank classifier gives an ordered list of classes associating each class with a integer that indicates its importance in the list; in the case of *K* classes, it is an integer k∈{1,2,⋯,K}. The output of a classifier $K_{k}$ is a vector of ranks attributed to *K* classes:(9)$$u_{k}\left(x\right)=r_{k}=\left(\begin{array}{c} r_{1k}\\ r_{2k}\\ \vdots \\ r_{Kk}, \end{array}\right),$$
and $r_{jk}=r_{k}\left(C_{j}\right)$ is the rank assigned to class $C_{j}$ by the classifier $K_{k}$. By convention, the smaller the rank assigned to a class, the more likely it is. In other words, $r_{ik}<r_{jk}$ if $K_{k}$ judges $C_{i}$ more likely than $C_{j}$. The vector $r^{\left(k\right)}$ is therefore a permutation of the first *K* integers. The matrix $R=\{r_{ik}\}$ represents the total order ranking of the *K* classes attributed by the *M* classifiers, i.e., $r_{ik}\neq r_{i^{\prime }k},\;\forall i^{\prime }\neq i$ [28]. In the following, for ease of writing, we will denote $r_{ik}=r_{i}^{\left(k\right)}$. Then
(10)$$R=\left(r_{1}\;r_{2}\;\cdots \;r_{M}\right)=\left(\begin{array}{cccc} r_{11} & r_{12} & \cdots & r_{1M}\\ r_{21} & r_{22} & \cdots & r_{2M}\\ \vdots & \vdots & \ddots & \vdots \\ r_{K1} & r_{K2} & \cdots & r_{KM}, \end{array}\right)$$
where $\left(r_{j1}\;r_{j2}\;\cdots \;r_{jM}\right)$ is the set of ranks assigned to class $C_{j}$ by the *M* classifiers.

The solution of a rank class combination problem is a total order ranking (TOR) $r^{*}$, given by a virtual classifier minimizing the disagreement of opinions between the *M* classifiers. The optimization problem is defined as follows: (11)$$r^{*}=\arg\min_{r}\sum_{k=1}^{M}f\left(r,r_{k}\right),\;s.t.\;r\in S_{K},$$
where $r_{k}$ is the rank distribution on the *K* classes proposed by the classifier $K_{k}$, $S_{K}$ is the symmetric group of the K! permutations [29], and $f:S_{K}\times S_{K}\to R^{+}$ is a metric on $S_{K}$. Solving Equation (Equation 11) is difficult due to the constraint $r\in S_{K}$. In the following subsections, the search for $r^{*}$ conducts to a linear optimization program with an exact solution that depends on the metric used, i.e., the disagreement distance or the Condorcet distance.

The choice of these metrics is motivated by a range of properties: (*i*) both have an intuitive and plausible interpretation as a number of pairwise choices, (*ii*) they provide the best possible description of the process of ranking classes as performed by a human, (*iii*) both have a number of appealing mathematical properties such as counting rather than measuring and providing a very good concordance indicator [30,31].

### 4.2. Total Order Ranking with Disagreement Distance

The disagreement between the rankings from classifiers $K_{k}$ and $K_{k^{\prime }}$ is measured by $f_{d}\left(r_{k},r_{k^{\prime }}\right)=\sum_{i=1}^{K}\mathrm{sgn}\left|r_{ik}-r_{ik^{\prime }}\right|$. The *k*th permutation $r_{k}$ can be represented by a permutation matrix $P^{\left(k\right)}=\left\{x_{ij}^{\left(k\right)}\right\},x_{ij}^{\left(k\right)}\in \left\{0,1\right\}$, with $x_{ij}^{\left(k\right)}=1$ if class *i* is positioned in place *j* and 0 otherwise (see Figure 2). Therefore, the constraint $r\in S_{K}$ in Equation (Equation 11) imposes $\sum_{j=1}^{K}r_{ij}^{*}=\sum_{i=1}^{K}r_{ij}^{*}=1,\forall i,j$. Let $\phi_{d}\left(r\right)=\sum_{k=1}^{M}f_{d}\left(r,r_{k}\right)={∥P,P^{\left(k\right)}∥}_{d}$ with tensor Einstein notation. Equation (Equation 11) can then be rewritten: (12)$$r^{*}=\arg\min_{r}\phi_{d}\left(r\right)=\arg\min_{r\in S_{K}}\sum_{k=1}^{M}\sum_{i=1}^{K}\mathrm{sgn}\left|r_{i}-r_{ik}\right|,$$
where $r_{i}$ denotes the rank of the *i*th candidate in the unknown ranking r. As r can be represented by its permutation matrix $P=\{x_{ij}\}$, it comes from the rewriting of $r_{i}=\sum_{j}jx_{ij}$ in Equation (Equation 12): (13)$$\phi_{d}\left(r\right)=\sum_{k=1}^{M}\sum_{i=1}^{K}\mathrm{sgn}\left|\sum_{j}jx_{ij}-r_{ik}\right|\;s.t.\;\sum_{j}x_{ij}=1,$$
which is equivalent to: (14)$$\phi_{d}\left(r\right)=\sum_{k=1}^{M}\sum_{i=1}^{K}\mathrm{sgn}\left|\sum_{j}\left(j-r_{ik}\right)x_{ij}\right|$$

Taking into account the summation on *j* and the fact that $x_{ij}$ only takes the value 1 once (and 0 elsewhere), only $\left(j-r_{ik}\right)$ corresponding to the value *j* for which $x_{ij}=1$ is considered. Then
(15)$$\phi_{d}\left(r\right)=\sum_{k=1}^{M}\sum_{i=1}^{K}\mathrm{sgn}(\sum_{j}^{K}|j-r_{ik}|x_{ij})=\sum_{k=1}^{M}\sum_{i=1}^{K}\sum_{j=1}^{K}\mathrm{sgn}(|j-r_{ik}\left|\right)x_{ij}.$$

Let us define by
(16)$$\kappa_{ij}\left(r\right)=\sum_{k=1}^{M}\mathrm{sgn}\left|j-r_{ik}\right|=\sum_{k=1}^{M}\left|x_{ij}-x_{ij}^{\left(k\right)}\right|$$
the cost of attributing the alternative *i* in position *j*. $\kappa_{ij}$ is also the number of classifiers that don’t position the alternative *i* in place *j*. $\kappa_{ij}\left(r\right)$ is equivalent to $m-\pi_{ij}$, where $\pi_{ij}$ is the number of classifiers who do position the alternative *i* in place *j*. Given that $|x_{ij}-x_{ij}^{\left(k\right)}|={\left(x_{ij}-x_{ij}^{\left(k\right)}\right)}^{2}$ because $|x_{ij}-x_{ij}^{\left(k\right)}|\in \left\{0,1\right\}$, we obtain
(17)$$\phi_{d}\left(r\right)=\frac{1}{2}\sum_{k=1}^{M}\sum_{i=1}^{K}\sum_{j=1}^{K}{\left(x_{ij}-x_{ij}^{\left(k\right)}\right)}^{2}=\sum_{k=1}^{M}\left(K-\sum_{i=1}^{K}\sum_{j=1}^{K}x_{ij}x_{ij}^{\left(k\right)}\right),$$
and then
(18)$$\phi_{d}\left(r\right)=\sum_{i=1}^{K}\sum_{i=1}^{K}\left(m-\sum_{k=1}^{M}x_{ij}^{\left(k\right)}\right)x_{ij}.$$

In Equation (Equation 18), considering that $\pi_{ij}=\sum_{k=1}^{M}x_{ij}^{\left(k\right)}$ is the number of classifiers that position class $C_{i}$ in place *j*, the linear objective function associated with Equation (Equation 12) is finally formulated as
(19)$$P^{*}=\arg\min_{P}\sum_{i=1}^{K}\sum_{j=1}^{K}\left(M-\pi_{ij}\right)x_{ij}\;s.t.\;\pi_{ij}=\sum_{k=1}^{K}x_{ij}^{\left(k\right)},\;\sum_{i=1}^{M}x_{ij}=\sum_{j=1}^{M}x_{ij}=1,\;\mathrm{and}\;x_{ij}\in \left\{0,1\right\},$$
constrained by $\sum_{i}^{K}\pi_{ij}=\sum_{j}^{K}\pi_{ij}=K$. The form to be minimized in Equation (Equation 19) recodes the classifier combination rule, which is reduced to solve an NP-hard binary linear programming problem (see [32] for some resolution strategies).

### 4.3. Total Order Ranking with Condorcet Distance

To define this distance, we define a new set of matrices $\left\{Y^{\left(1\right)},\cdots ,Y^{\left(m\right)}\right\}$, where $Y_{ij}^{\left(k\right)}=\left\{y_{ij}\right\}={𝟙}_{i<j}$ is put for the indicator matrix of classifier $K_{k}$ with the convention $y_{ij}^{\left(k\right)}=1$ if the rank of class $C_{i}$ is less than that of class $C_{j}$ and 0 otherwise (see Figure 3).

Using the tables $Y^{\left(k\right)}$ as in Section 4.2
(20)$$f_{C}\left(r_{k},r_{k^{\prime }}\right)=f\left(Y^{\left(k\right)},Y^{\left(k^{\prime }\right)}\right)=\frac{1}{2}\sum_{i}^{K}\sum_{j}^{K}\left|y_{ij}^{\left(k\right)}-y_{ij}^{\left(k^{\prime }\right)}\right|,\;k,k^{\prime }=1,\cdots ,M,$$
which can be simplified as follows in the case of total order: (21)$$f_{C}\left(r_{k},r_{k^{\prime }}\right)=\frac{1}{2}\sum_{i}^{K}\sum_{j}^{K}{\left(y_{ik}^{\left(k\right)}-y_{ik}^{\left(k^{\prime }\right)}\right)}^{2}=\sum_{i}\sum_{j}y_{ij}^{\left(k\right)}y_{ji}^{\left(k^{\prime }\right)}.$$

As ${\left(y_{ij}^{\left(k\right)}\right)}^{2}=y_{ij}^{\left(k\right)}=0$ or 1, the consensus function associated with the Condorcet distance is given by
(22)$$\phi_{C}\left(r\right)=\frac{1}{2}\left[\sum_{i=1}^{K}\sum_{j=1}^{K}My_{ij}+\sum_{i=1}^{K}\sum_{j=1}^{K}\left(\sum_{k=1}^{M}y_{ij}\right)-2\sum_{i=1}^{K}\sum_{j=1}^{K}y_{ij}\sum_{k=1}^{M}y_{ij}^{\left(k\right)}\right].$$

Let $\delta_{ij}=\sum_{k=1}^{K}y_{ij}^{\left(k\right)}$ be the total number of classifiers preferring class $C_{i}$ to $C_{j}$. Defining $\Delta =\{\delta_{ij}\}$ as a matrix summing the *M* matrices $Y^{\left(k\right)}$ associated with the rankings $r_{k}$ of the classifier $K_{k}$ allows us to rewrite $\phi_{C}$ as
(23)$$\phi_{C}\left(r\right)=\frac{1}{2}\left[\sum_{i=1}^{K}\sum_{j=1}^{K}My_{ij}+\sum_{i=1}^{K}\sum_{j=1}^{K}\delta_{ij}-2\sum_{i=1}^{K}\sum_{j=1}^{K}\delta_{ij}y_{ij}\right].$$

As r defines a total order, $\sum_{i=1}^{K}\sum_{j=1}^{K}y_{ij}=\frac{K(K-1)}{2}$ and $\sum_{i=1}^{K}\sum_{j=1}^{K}\delta_{ij}<M\frac{K(K-1)}{2}$.

Let $\theta =\frac{1}{2}\left(M\frac{K(K-1)}{2}+\sum_{i=1}^{K}\sum_{j=1}^{K}\delta_{ij}\right)$. Then theta is constant and $\phi_{C}\left(r\right)$ is
(24)$$\phi_{C}\left(r\right)=\theta -\sum_{i=1}^{K}\sum_{j=1}^{K}\delta_{ij}y_{ij}.$$

Finally, the search for an optimal rank classifier combination conducts to the following binary linear program:(25)$$\begin{array}{c} \max_{Y}\sum_{i=1}^{K}\sum_{j=1}^{K}\delta_{ij}y_{ij}\;s.t.\;\delta_{ij}=\sum_{k=1}^{K}y_{ij}^{\left(k\right)},\;y_{ij}+y_{ji}=1,i<j,\;y_{ii}=0\forall i\\ \;y_{ij}+y_{ji}-y_{ik}\leq 1,\forall i\neq j\neq k,\;y_{ij}\in \left\{0,1\right\}. \end{array}$$

From a machine learning perspective, solving Equations (Equation 19) and (Equation 25) provides deterministic matrix solutions $P^{*}$ and $Y^{*}$, respectively, from which r* is easily reconstructed, but these solutions are not necessarily identical [28].

**Example** **3**(Classifier ensemble aggregation rule)**.**
*The problem selected to illustrate our theory is that of combining four classifiers for recognizing handwritten digits 0 to 9. Binary images from the MNIST database are used [33]. The four classifiers are tested on a sample and proposed rankings are collected in Table 2.*The two rankings are concordant except for the predictions for digits 5 and 6.

## 5. Classifier Ensemble Information Measure

Since sgn|x|≤|x|, then
(26)$$f_{d}\left(r_{k},r_{k^{\prime }}\right)=\sum_{i=1}^{K}\mathrm{sgn}|r_{ik}-r_{ik^{\prime }}|\leq \sum_{i=1}^{K}\left|r_{ik}-r_{ik^{\prime }}\right|.$$
If $r_{ik}=K-\sum_{i=1}^{K}y_{ij}^{\left(k\right)}$, then from Equation (Equation 26),
(27)$$\sum_{i=1}^{K}\left|r_{ik}-r_{ik^{\prime }}\right|=\sum_{i}^{K}\left|\sum_{j}^{K}\left(y_{ij}^{\left(k\right)}-y_{ij}^{\left(k^{\prime }\right)}\right)\right|\leq \sum_{i}^{K}\sum_{j}^{K}\left|y_{ij}^{\left(k\right)}-y_{ij}^{\left(k^{\prime }\right)}\right|=2f_{C}\left(r_{k},r_{k^{\prime }}\right).$$
In summary, $f_{d}\left(r_{k},r_{k^{\prime }}\right)\leq f_{C}\left(r_{k},r_{k^{\prime }}\right)$, which means that $f_{C}$ is more uncertain than $f_{D}$ and could be preferred for a classifier ensemble agreement. The question is, how precisely can we measure this voting conjunction?

Section 4.3 introduced a matrix representation of the information. By summing for all the tables $Y^{\left(k\right)}$, one obtains the matrix Δ defined previously. If we arrange the classifiers according to a permutation order Σ=(σ(1),σ(2),⋯,σ(K)), Δ can be represented from matrix Δ(Σ) obtained by the permutation of rows and columns.

The objective function to minimized is given in the general case by: (28)$$F_{C}r=\theta -(\mathrm{sum}\;\mathrm{of}\;\mathrm{the}\;\mathrm{elements}\;\mathrm{of}\;\mathrm{the}\;\mathrm{upper}\;\mathrm{triangular}\;\mathrm{part}\;\mathrm{of}\;\mathrm{the}\;\mathrm{matrix}),$$
and as follows in the case of total orders: (29)$$F_{C}\left(r\right)=(\mathrm{sum}\;\mathrm{of}\;\mathrm{the}\;\mathrm{elements}\;\mathrm{of}\;\mathrm{the}\;\mathrm{lower}\;\mathrm{triangular}\;\mathrm{part}\;\mathrm{of}\;\mathrm{the}\;\mathrm{matrix}).$$
A measure of classifier ensemble agreement is a coefficient between 0 and 1 measuring the intensity of the link between the set of classifier votes. The closer its value is to 1, the more the opinions of the classifiers are in agreement. Conversely, the closer their value is to 0, the greater the disagreement between the votes. Here, we give the coefficients of concordance for the two metrics.

### 5.1. Disagreement Distance

**Theorem** **1**(Conjunction coefficient interval for the disagreement metric)**.**
*Let ${\left\{K_{i}\right\}}_{i=1}^{M}$ be an ensemble of conditionally independent classifiers voting on K classes. Then, the interval of variation of the conjunction coefficient $I_{d}$ is [0,1].*

See Appendix A for the proof.

### 5.2. Condorcet Distance

If *M* classifiers vote on *K* classes with pairing order comparison matrices $Y^{\left(k\right)}$, the sum of which makes it possible to obtain $\Delta =\{\delta_{ij}\}$ with $\delta_{ij}=\sum_{k=1}^{K}y_{ij}^{\left(k\right)}$, as defined in Section 4.3, the conjunction coefficient is defined as
(30)$$I_{C}=4\frac{\sum_{j=1}^{K}\sum_{j=1}^{K}\delta_{ij}\left(\delta_{ij}-1\right)}{M(M-1)K(K-1)}-1.$$

**Theorem** **2**(Conjunction coefficient interval for the Condorcet metric)**.**
*Let ${\left\{K_{i}\right\}}_{i=1}^{M}$ be an ensemble of conditionally independent classifiers voting on K classes. Then the interval of variation of the conjunction coefficient $I_{C}$ defined by (Equation 30) is*
(31)$$I_{C}\in \left\{\begin{array}{cc} \left[-\frac{1}{M};1\right] & \;if\;M\;is\;even,\\ \frac{1}{M-1};1] & \;otherwise. \end{array}\right.$$

See Appendix B for the proof.

## 6. Experiments

### 6.1. The Detection of Cervical Cancer

Many studies have shown evidence that cervical cancer may be imputed to a subset of DNA viruses called *human papillomavirus* (HPV) (referred to as risky patients)that infect cutaneous and mucosal epithelia, and in which acute infection causes benign cutaneous lesions [34,35]. Some of these viruses infect the genital tract and cause malignant tumors, which are most commonly located in the cervix. Even though most of these infections are controlled by the immune system, some remain persistent and are ascribed to different types of cancers and particularly, to cervical cancer. In 2016, cervical cancer represented the 12th most lethal female cancer in the European Union, accounting for 13500 deaths a year and 30400 new cases a year. Therefore, cervical cancer screening still continues to play a critical role in the control of cervical cancer. However, the screening of a smear is nowadays mostly made manually: a pathologist inspects each cell of a smear with a microscope to check if it is atypical or not. Consequently, human error is always possible, and in particular, mistakenly diagnosing atypical cells as normal. This situation can occur because of the practitioner’s fatigue or a lack of experience or concentration. In addition, diagnosis is also linked to the preparation of cells, and in some situations, atypical cells can be partially hidden by others, which makes their interpretation or classification difficult. In addition, the presence of atypical cells in the entire studied population is very uncommon (up to 1‰) which makes the detection task even more difficult. Therefore, an error is easily possible. This could have irreversible effects on the evolution of the cancer and can impact treatment. The introduction of an automatic procedure, able to point out the pathological cells, would both help the practitioner in his diagnosis and improve or strengthen it.

Depending on the morphology of the nuclei of the cells, the diagnosis varies: if a nucleus is considered normal and all of the cells removed have the same diagnosis, then the cervix is considered normal. On the other hand, if a nucleus is considered abnormal, the diagnosis is not automatically associated with a risky smear.

We propose to test our classifier combination strategy to cluster cells into three different classes (normal cells, atypical cells, and debris) using a certain number of classifiers.

### 6.2. The Dataset

The cytological dataset is constituted of smear images from 14 different women. They generally comprise more than one hundred cells characterized by 42 morphological or textural variables. Nine showed a negative hpv test and the other five, a positive test. In addition, few observations were labeled by an expert who pointed out some atypical cells and noisy objects. The dataset is presented in detail in Table 3. Among the most recurrent patterns of abnormal cells are nucleus regularity or a swollen aspect, nucleus size, important optical density, number of nucleoli, high core/cytoplasm ratio, ratio of minimum/maximum width of the nucleus, etc.

The images were colored with Papanicolaou stain, which is the most widely used reference color for the screening of cervical cancers; it makes it possible to distinguish the different nuclei, which are colored in blue, the mother cells in dark purple to black, and the keratinized and squamous epithelium. The images were then segmented into thumbnail images of 16×16 pixels which correspond a priori to objects. Most of the time, these objects are nuclei, but they may sometimes be non-identified objects that we call “noise”. Indeed, they can correspond, for example, to a poor segmentation, a superimposed nuclei, etc.

A few observations were labeled by an expert who pointed out some atypical cells and noisy objects. The fact that a nucleus has one of these characteristics does not always imply its malignancy. In fact, a cell can have a singular morphology but not be infected, and others may present abnormalities that correspond to pre-cancerous lesions such as dysplastic cells and in situ carcinomas or to cancerous cells. Figure 4a shows a cluster of abnormal cells (with large nuclei) that are not yet cancerous, because of their low density, unlike Figure 4b, where one can observe a set of abnormal cells with dense nuclei.

Table 4 summarizes the characteristics of the dataset. First, the observed data come from samples of 14 different smears, which supposes the existence of inter-individual variability (confirmed by tests of variance between the hpv negatives, the hpv positives, or between the two types of population; the 5% risk threshold tests rejected the assumption of equality of means for all variables). However, it is possible that this variability is simply relative to the studied dataset, in the sense that the study was done on a small number of smear samples. This assumption remains to be verified on larger databases. It can also be noted in Table 3 that the known population of “abnormal” cells remains very low in proportion to the other classes, and, in contrast, the recognized “default/waste” class represents more than 15% of the data. The low proportion of the target class and the heterogeneity of the debris present obstacles for clustering. This means that, among the cells belonging to risky patient smears, there exists a non-null risk that some nuclei are atypical. Iin practice, this proportion is usually very low (0.1% to 5%).

From this image segmentation, morphological and photometric features are extracted and computed. In total, the studied dataset has 3857 cell samples belonging to 14 different smears and consists of 42 variables: variables 1 to 19 represent morphological variables, and the rest corresponds to textural and photometric characters. The channel of treatments from the smear image to the dataset is reported in Figure 5.

Each smear was pre-processed according to a standardized protocol: cell collection, spreading a thin layer on slides, and the staining of these slides. Each slide was then scanned, segmented cell by cell, and finally, underwent an extraction of 42 morphological and textural characteristics.

### 6.3. Experimental Protocol

Two-layer multilayer perceptrons (MLPs) were chosen as classifiers to produce the desired outputs, which were ordered to produce the ranks. Each multilayer perceptron (MLP) contains 42 input units, 10 hidden units, and 3 output units. Training was achieved using a learning rate of 0.1 and a momentum of 0.9 for two epochs on the training set. We deliberately trained the MLPs without optimization of a validation set. It is important to stress that the training set for the classifier was not the same set as the test set, the ensure that the experiments would be unbiased. The best results obtained for an mlp were a classification error rate of 0.159±0.022 and a false positive rate (FPR) (or false alarm ratio) of 0.133±0.050. The fpr is the number of false positives divided by the total number of negatives *N*, i.e., FP/N. The false negative rate (FNR) is the number of false negatives divided by the number of real positive cases in the data, i.e., FN/P. In practice, this is a test result that indicates that a condition does not hold, while in fact it does.

In order to assess the efficiency of the rank classifier combination algorithms, error rates were computed from a certain percentage of nuclei whose labels were known. This represented 70% of the observations in a subsample, as we took into account the 20 labeled atypical nuclei randomly selected, and we also assumed that those coming from control patients (120) were all normal nuclei. We proceeded in the same manner to compute the fpr which stands for the percentage of actual atypical nuclei mis-classified.

In Table 5 and in Figure 6a, we report the classification error rate computed from the data with known labels and its corresponding fpr, for the two procedures. First of all, we can observe that the Condorcet combination rule shows the best performances in terms of classification error rate and fpr (see also Figure 6c). Indeed, only 4.28% of cells are mis-classified, whereas the disagreement combination rule has a mis-classification rate of 4.84% in the best case, with 765 classifiers. The main conclusion is that the success ratio is strongly improved when combining classifiers. However, it is disappointing to see that the Condorcet algorithm results in a significant number of false negatives (pathological cells classified as normal ones); the fnr also remains relatively high, around 10%, in many simulations. Indeed, the classification risk is not symmetric here: the detection of pathological cells activates the decision for treatment, and their absence implies an absence of treatment.

We compared the clustering partition obtained by the three competitors: sparse *k*-means (SkM) proposed by Witten and Tibshirani [36,37], general sparse multi-class linear discriminant analysis (GSM-LDA) [38], and sparse EM (sEM)by Zhong et al. [39]. First, we can observe that among these algorithms, the sEM shows the best performance in terms of clustering accuracy. Only 9% of observations are mis-classified, on average, whereas the GSM-LDA algorithm has a mis-classification rate of 15.9%, and the SkM algorithm mis-classifies 19.2% of nuclei. However, the sparse approaches provide a better clustering results from a medical point of view since the results can be interpreted conversely to the LDA-type algorithm, for which the fitted discriminative axis is a linear combination of the original variables. Therefore, SkM and sEM provide information which can be interpreted to better understand both the data and the phenomenon.

The rank classifier combination provides the best classification results. We can observe that the global clustering error rates are considerably reduced (Table 6). Indeed, the best error rate reaches 4.28% with 907 classifiers and a conjunction coefficient of 96.6%.

## 7. Conclusions and Future Research

In this paper, we show that an exact optimal combination rule for a rank classifier ensemble can be computed as the solution to a binary linear programming problem. This rule can be seen as a total order ranking attributed to *K* classes by a virtual voter resuming the points of view of *M* voters. One could also stand the dual problem of the previous one, i.e., is there a distribution of marks or values that could have been attributed to a virtual class *C* by the *m* voters? The first problem is related to the idea of aggregating points of view, the second with the idea of summarizing profiles.

We compared disagreement and Condorcet metrics, making it possible to quantify the consensus between the classifiers with a conjunction coefficient. The optimal rankings are not the same, i.e., the solution depends of the metric used. But they have shown their efficiency, in addition to the appealing property of being deterministic algorithms: they improve the classification results and ease the interpretation and the understanding of the results. Another point worth mentioning is the theoretical capability of handling the reject option. A weak point of this technique is that it treats all classifiers equally and does not take into account individual classifier capabilities. This disadvantage can be reduced to a certain degree by applying weights. The weights can be different for every classifier, which in turn requires additional training. This idea deserves to be further explored.

The role of variable selection appears to be significant, as it enables the improvement of both the clustering partition and the modeling of the atypical cells in the cancer detection smear (see Figure 5). In the future, we propose including a rule to rank the selected features and to investigate how the number and nature of classifiers influence the results of the rank classifier combination.

## Figures and Tables

**Figure 1 entropy-21-00440-f001:**
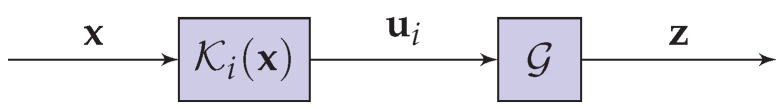
General framework for classifier combination. The classifier $K_{i}\left(x\right)$ produces output vector $u_{i}$. Finally, from $u_{i}$ the combination function produces a final decision vector z.

**Figure 2 entropy-21-00440-f002:**
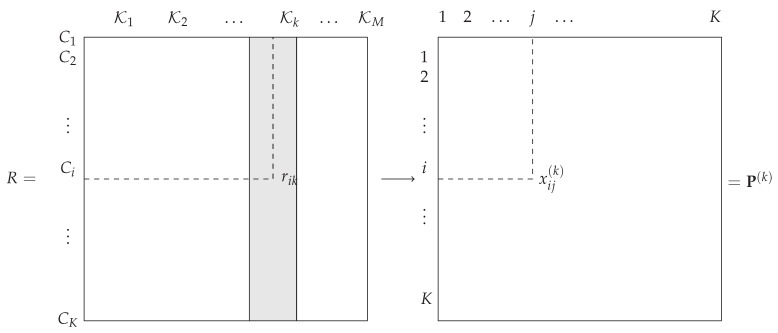
Permutation matrix put for the ranking of classifier $K_{k}$.

**Figure 3 entropy-21-00440-f003:**
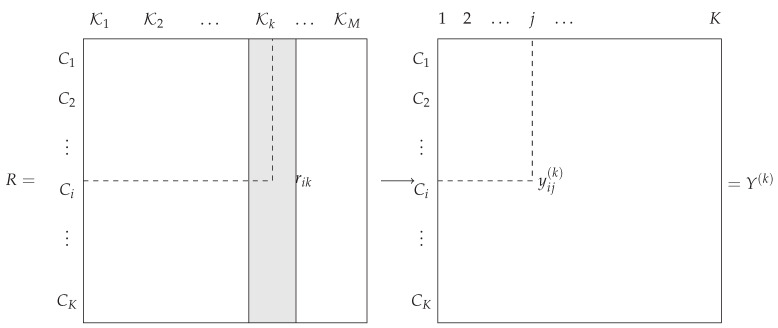
Condorcet matrices.

**Figure 4 entropy-21-00440-f004:**
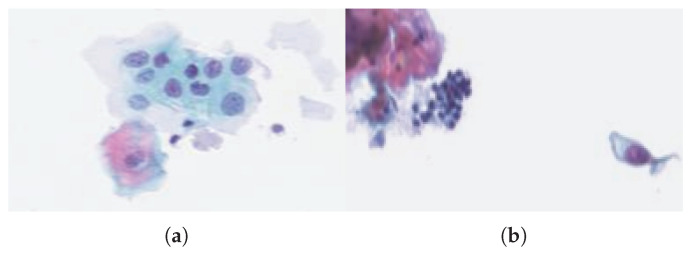
Images of cervical cells colored with Papanicolaou stain. (**a**) Clumps of abnormal cells with large nuclei. (**b**) Abnormal cells with dense nuclei.

**Figure 5 entropy-21-00440-f005:**
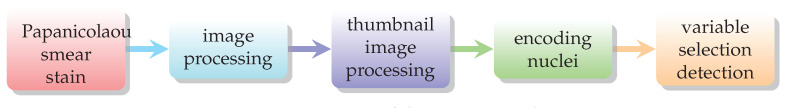
Overview of the processing chain.

**Figure 6 entropy-21-00440-f006:**
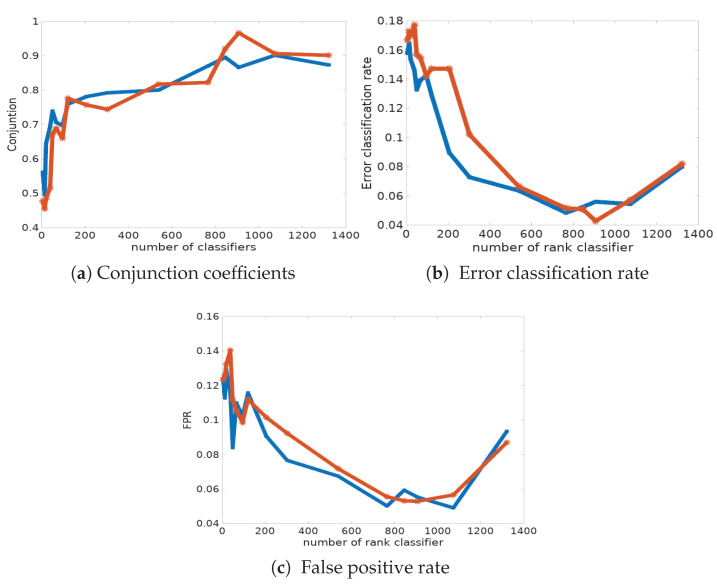
Graphic representations of the classification results for disagreement (blue) and Condorcet (red) distances.

**Table 1 entropy-21-00440-t001:** Confusion matrix of a classifier $K_{i}$ used to estimate $p(U_{ik}|C_{j})$ in the Bayesian approach. $U_{i}=R_{j}$ denotes the classifier decision on class being ranked *j*th.

		Predicted Classes					
		$R_{1}$	…	$R_{j}$	…	$R_{K}$	
True classes	$C_{1}$	$n_{11}$	⋯	$n_{1j}$	⋯	$n_{1K}$	$n_{1\;\cdot \;}$
	⋮	⋮	⋱	⋮		⋮	⋮
	$C_{j}$	$n_{j1}$	⋯	$n_{jj}$	⋯	$n_{jK}$	$n_{j\;\cdot \;}$
	⋮	⋮		⋮		⋮	⋮
	$C_{K}$	$n_{K1}$	⋯	$n_{Kj}$	⋯	$n_{KK}$	$n_{K\;\cdot \;}$
		$n_{\;\cdot \;1}$	⋯	$n_{\;\cdot \;j}$	⋯	$n_{\;\cdot \;K}$	

**Table 2 entropy-21-00440-t002:** Proposed rank classifier combination using disagreement and Condorcet distances.

Digits	Classifier Ranks				Proposed Rank	
	$K_{1}$	$K_{2}$	$K_{3}$	$K_{4}$	**Disag.**	**Condorcet**
0	1	4	3	10	3	3
1	2	2	1	2	2	2
2	3	1	2	1	1	1
3	4	6	4	3	4	4
4	5	5	7	5	5	6
5	6	3	6	4	6	5
6	7	8	5	6	7	7
7	8	7	8	9	8	8
8	9	10	10	8	10	10
9	10	9	9	7	9	9

**Table 3 entropy-21-00440-t003:** Dataset characteristics.

HPV Test	Total Number of Cells	Number (or %) of			
		– Debris – – Cancer –			
positive	405	78	*(0.19)*	49	*(0.12)*
positive	114	19	*(0.17)*	8	*(0.07)*
positive	206	31	*(0.15)*	13	*(0.06)*
positive	448	30	*(0.06)*	2	*(0.004)*
positive	519	70	*(0.13)*	33	(0.06)
negative	137	13	*(0.09)*	–	–
negative	76	5	*(0.06)*	–	–
negative	211	84	*(0.39)*	–	–
negative	251	31	*(0.12)*	–	–
negative	251	52	*(0.20)*	–	–
negative	257	40	*(0.15)*	–	–
negative	223	24	*(0.11)*	–	–
negative	691	155	*(0.22)*	–	–
negative	67	23	*(0.24)*	–	–
Total	3857	655	*(0.17)*	105	*(0.02)*

**Table 4 entropy-21-00440-t004:** Overview of the studied dataset.

	No. of Patients	No. of Nuclei	No./Yype of Data
control patients	9	2165	427/noisy objects
risky patients	5	1692	105/atypical nuclei
	–	–	228 / noisy objects
Total	14	3857	760 objects

**Table 5 entropy-21-00440-t005:** Classification results with disagreement and Condorcet combination rules using a set of *M* classifiers (with 4≤M≤1321).

Disagreement Distance					Condorcet Distance				
$I_{d}$	M	Error Rate	FPR	FNR	$I_{C}$	M	Error Rate	FPR	FNR
0.873	1321	0.0800	0.0934	0.0644	0.901	1321	0.0820	0.0870	0.0777
0.901	1073	0.0544	**0.0491**	0.0576	0.906	1073	0.0572	0.0566	0.0561
0.866	907	0.0560	0.0553	0.0562	0.966	907	**0.0428**	0.0529	**0.0313**
0.895	845	0.0524	0.0593	**0.0464**	0.920	845	0.0508	**0.0532**	0.0465
0.870	765	**0.0484**	0.0502	0.0500	0.822	765	0.0516	0.0555	0.0493
0.800	538	0.0636	0.0675	0.0600	0.817	538	0.0664	0.0718	0.0592
0.792	302	0.0728	0.0766	0.0710	0.744	302	0.1020	0.0923	0.1126
0.781	205	0.0896	0.0906	0.0864	0.757	205	0.1472	0.1015	0.1968
0.759	120	0.1292	0.1157	0.1439	0.776	120	0.1472	0.1118	0.1846
0.697	95	0.1424	0.1023	0.1809	0.660	95	0.1424	0.0985	0.1888
0.706	66	0.1392	0.1098	0.1667	0.689	66	0.1548	0.1055	0.2090
0.739	49	0.1328	0.0840	0.1854	0.672	49	0.1568	0.1117	0.2067
0.694	38	0.1460	0.1152	0.1770	0.516	38	0.1772	0.1404	0.2233
0.643	19	0.1540	0.1294	0.1801	0.484	19	0.1696	0.1323	0.2032
0.496	13	0.1644	0.1127	0.2224	0.455	13	0.1728	0.1261	0.2248
0.561	4	0.1580	0.1238	0.1962	0.477	4	0.1668	0.1234	0.2130

**Table 6 entropy-21-00440-t006:** Results obtained for the sparse *k*-means (SkM), general sparse multi-class linear discriminant analysis (GSM-LDA), and sparse EM (sEM) algorithms: Average and standard error of clustering error rate, false positive rate fpr, and false negative rate fnr on 20 simulations.

Algorithm	Error Rate	FPR	FNR
skm [36]	0.192±0.016	0.205±0.044	0.165±0.084
gsm [38]	0.159±0.022	0.133±0.050	0.118±0.099
sem [39]	0.090±0.047	0.077±0.022	0.062±0.061

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
