# Peer review of "M*-ary Rank Classifier Combination: A Binary Linear Programming Problem"

_entropy, 2019, doi:10.3390/e21050440_

Round 1
Reviewer 1 Report
In this manuscript,the authors propose a method for ensembling classifiers. This work is significant, and the manuscript is well organized. Proposals are interesting and well presented, obtaining satisfactory experimental results. I think the manuscript can be accepted after addressing following issues:
1. The indentation of each paragraph is confused, please adjust it.
2. In the experiments, could the authors compare the proposed method with other methods for classifier combination?
3. Could the authors provide some possible future works of the proposed method which can be done in further research?
Author Response
Dear Reviewer,
the paper has been fully revised since the past days. These changes concerned typos, rewriting of introduction and conclusion, modifications in the notations, corrections in every sections of the text. 2 tables were added (table 4 the overview of the dataset, table 6 that gives the results obtain with the 3 competitors), etc.
An other major change is the title of the paper "M-ARY RANK CLASSIFIER COMBINATION: A BINARY LINEAR PROGRAMMING PROBLEM" that highlight better the content of the article.
Considering yours remarks :
1. The indentation of each paragraph is confused, please adjust it.
indeed, we cancel in the .tex file an inappropriate \noindent instruction that fix this indentation issue.
2. In the experiments, could the authors compare the proposed method with other methods for classifier combination?
The comparisons have been forecasted since the beginning of this work. We wanted initialy to include them in a second future paper rather to include then in this paper. The competitors are sparse k-means, sparese EM and GSM-LDA. The second one performs quite good on our data.
You are write it makes sens e to include these comparisons right now.
The results in table 6 are directly comparable with the 2 previous tables.
3. Could the authors provide some possible future works of the proposed method which can be done in further research?
thanks for this remark. We include in the conclusion highligts on future works:
one concern the input/feature selection of the classifier that could be chosen using our rank-order classifier combination rule, the second point to investigate is a question : how the number and nature of classifiers influence the results of the rank classifier combination?"
Reviewer 2 Report
- complete the abstract by giving an overview or highlight about results obtained or the importance of the results. --Abstract too short!
- define K in eq 2
-line 117, p.2 typo: are rather than ar
- line 201 -p.11, typo: opposite rather oppose
- line 253 , p. 13, typo: activates rather than activate
- 7. conclusion : line 258 - a family .... an optional Line 266 -- rules ...show their efficiency. In addition...
- The last sentence : This option ...erroneous decision specially in the medical domain.
Author Response
Dear Reviewer,
the paper has been fully revisited since the past days. These changes concerned typos, rewriting of introduction and conclusion, modifications in the notations, corrections in every sections of the text. 2 tables were added (table 4 the overview of the dataset, table 6 that gives the results obtain with the 3 competitors), etc.
An other major change is the title of the paper "M-ARY RANK CLASSIFIER COMBINATION: A BINARY LINEAR PROGRAMMING PROBLEM" that highlight better the content of the article.
We took in consideration also the following remarks :
- complete the abstract by giving an overview or highlight about results obtained or the importance of the results. --Abstract too short!
Abstract
has been extended and highlights 2 main results: the classification
rate is strongly improved by combining rank classifiers and the fact
that the combination rule is the solution of a binary linear program.
- define K in eq 2
Defined previously in the notations paragraph, before the 2nd section
-line 117, p.2 typo: are rather than ar
corrected
- line 201 -p.11, typo: opposite rather oppose
corrected
- line 253 , p. 13, typo: activates rather than activate
corrected
- 7. conclusion : line 258 - a family .... an optional Line 266 -- rules ...show their efficiency. In addition...
- The last sentence : This option ...erroneous decision specially in the medical domain.
we rewrote completely the conclusion to highlight future works and put forward the main ideas behind the article.
Round 2
Reviewer 1 Report
The authors have revised the manuscript to address my suggestion and concern, and I am satisfied with the revision. I think this manuscript can be accepted.